# "For More Diversity, Better Taste and My Own Health" Exploring Organic Consumers' Purchasing Motives for Heirloom Vegetable Varieties

Josephine Lauterbach * and Christina Bantle

Faculty of Landscape Management and Nature Conservation, Eberswalde University for Sustainable Development, Schicklerstr. 5, 16225 Eberswalde, Germany; christina.bantle@hnee.de
* Correspondence: josephine.lauterbach@hnee.de

**Abstract:** Agrobiodiversity is the foundation of our ecosystems and food supply. However, agrobiodiversity is declining rapidly. A prominent strategy to safeguard endangered varieties, an important component of agrobiodiversity, is their cultivation and preservation in their natural environments. In order to make the cultivation of these varieties attractive to farmers, a functioning value chain and communication concepts for these goods have to be developed. Using heirloom vegetable varieties as an example, we examine existing communication approaches for endangered varieties and evaluate their suitability to transport their added value to organic consumers. We furthermore examine organic consumers' purchasing motives to buy heirloom vegetable varieties. We collected data in three focus group discussions in Berlin (Germany) in 2018. This exploratory study shows that existing communication approaches for heirloom vegetable varieties strongly appeal to altruistic and biospheric purchasing motives. However, our results suggest that egoistic and hedonic purchasing motives are just as important to organic consumers. Hence, existing communication approaches for heirloom vegetable varieties, including the "Red List of Endangered Local Crops", are not entirely suitable to communicate the added value of biodiversity-enhancing products to consumers. Based on these results, we will develop a holistic communication scheme for heirloom vegetable varieties for organic supermarkets and further distribution channels in Germany.

**Keywords:** communication; agrobiodiversity; heirloom vegetable varieties; red list; focus groups

## 1. Introduction

According to the Convention on Biological Diversity, agrobiodiversity can be defined as a "broad term that includes all components of biological diversity of relevance to food and agriculture [ . . . ] the variety and variability of animals, plants and micro-organisms, at the genetic, species and ecosystem levels, which are necessary to sustain key functions of the agro-ecosystem, its structure and processes" [1] (COP decision V/5, appendix). Thus, agrobiodiversity is the foundation for the resilience of our agro-food systems [2]. One crucial component of agrobiodiversity is plant genetic resources.

Plant genetic resources are "the foundation of food production, and the biological basis for food security, livelihoods and economic development" [3]. Their current loss poses a threat to food security, local livelihoods, and the adaptability of ecosystems to changing environmental conditions [4,5].

To preserve the genetic potential of plant genetic resources, they need to be protected. One prominent strategy to protect plant genetic resources is the *on-farm* conservation of endangered crops and varieties, meaning the active cultivation of endangered plants in their natural environments.

Yet, this strategy is not attractive for farmers because the cultivation of endangered crops is often labour intensive, and those plants are often not adapted to a modern agricultural system [6]. To make the cultivation of endangered crops attractive for horticulturists,

a valorisation strategy must be established. This includes creating a sufficient demand and a functioning value chain for these products [7,8].

On a large scale, such a valorisation strategy is lacking in Germany. Biodiversity-enhancing products, such as landraces, heirloom varieties, and underutilized crops, are currently almost exclusively sold via direct marketing, e.g., in farm shops or at market stalls [9–11]. Although organic supermarkets have gained a considerable market share in Germany over the last 10 years [12], experiences from indirect marketing channels, especially organic supermarkets, are rare. Consequently, their potential for the sale and thus conservation of heirloom varieties has not yet been utilised.

A major challenge for indirect marketing channels lies in the lack of personal communication between the producer and consumer [13]. Biodiversity-enhancing products are often in need of further explanation. They can be classified as 'credence good,' as their added value, such as the conservation of agrobiodiversity, cannot be assessed by consumers directly [14]. Therefore, effective and target-group-oriented communication tools for retailers are necessary [11,15].

Hence, we seek to gain insights into purchasing motives for heirloom vegetable varieties and try to evaluate whether existing communication approaches from conservation organisations and marketing initiatives are suitable to transport the added value of heirloom vegetable varieties to consumers. We focus on organic consumers as a first suitable target group.

In this study, we address the following research questions:

1.　Which attitudes do organic consumers express towards heirloom vegetable varieties?
2.　What are organic consumers' driving purchasing motives and information requests for heirloom vegetable varieties?
3.　How do organic consumers assess existing communication approaches for heirloom vegetable varieties?

We use the results to develop target-group-oriented and scientifically sound recommendations for a holistic communication approach to be used by organic retailers selling biodiversity-enhancing products in Germany. The results are also valuable for other countries with well-developed organic markets [16] in which a further differentiation within the market (e.g., between different varieties) may represent a marketing advantage.

With the developed communication approach, heirloom vegetable varieties will be accessible to a larger group of potential buyers, thus enabling consumers to actively support biodiversity conservation in their everyday grocery purchases.

## 2. Literature Review

### 2.1. Importance of Plant Genetic Resources

Diversity of plant genetic resources is a crucial component to help the agricultural system adapt to future challenges, such as climate change or pests. It increases genetic potential with which breeders can react to new social and ecological challenges. Agrobiodiversity thus makes an important contribution to food security, the resilience of ecosystems, the absorption of abiotic stress factors, and the adaptation of agricultural systems to climate change. It provides a pool of genetic resources needed to adapt to changes in natural systems [11,17–19].

Consumers also benefit from a wide variety of species and varieties used. New products can be developed, e.g., with more valuable ingredients or more diverse sensory properties (e.g., appearance, taste) [11].

Despite these advantages, there is currently a significant loss of plant genetic resources used in agricultural systems. This is due to the standardisation of cultivation systems, the use of a few high-performance varieties, and consolidation in the breeding sector [2,19,20].

### 2.2. Heirloom Varieties

Plant genetic resources comprise the genetic variety between and within species. This includes land-races, which are also known as farmers' varieties or underutilized crops [21,22]. Heirloom vegetable varieties are a vital part of those resources. They are

currently not in commercial use, as they have been forgotten, lost their commercial importance, or have been replaced by modern varieties. However, they possess a potential to strengthen the resilience of agricultural systems and to diversify and improve the existing value chain in terms of sustainability and resilience [22,23].

The German Ministry for Agriculture categorizes heirloom varieties as follows:

- Lost varieties: mentioned in historic sources, but not available in seed banks anymore and without variety approval within the EU;
- Traditional varieties: mentioned in historic sources and currently with variety approval;
- 'Red List' varieties: mentioned in historic sources and available in seed banks, but without variety approval within the EU [24].

The "Red List of Endangered Local Crops", which was last updated in 2018, contains over 2600 crop traits and varieties. They can be classified as indigenous (e.g., old German land variety), endangered (low to no current occurrence in situ/on-farm) or significant (e.g., potential for use by consumers or breeders). However, they no longer have variety protection or variety approval within the EU. Similar to the "Red List of Endangered Species" in nature conservation, the "Red List of Endangered Local Crops" is used to draw attention to the endangered status of useful plants and provides a higher incentive to conserve these species [25,26]. Hence, we include the "Red List" as a potential communication approach in this study.

Plant diversity can be protected and preserved either in situ/on-farm or ex-situ.

In ex-situ conservation, plant genetic resources are stored in collections (gene banks) for conservation purposes. The genetic material is documented and characterised. The material is kept in its current status quo and is not further developed.

In-situ conservation is defined as the conservation of ecosystems and cultivated plants in their natural surroundings. On-farm conservation, as a special form of in situ conservation, is defined as the preservation of locally adopted regional varieties in the surroundings in which they have been developed. Hence, in on-farm conservation, the material is not merely stored but can be used to diversify the variety of cultivated species, human nutrition, or to generate additional income for farmers [27].

*2.3. Consumer Attitudes towards Heirloom Varieties*

As this study investigates consumer attitudes towards heirloom vegetable varieties, in the following section we provide a brief overview on the existing literature on this topic, mainly focusing on Germany.

Several studies [9,28,29] provide a general overview of consumer attitudes towards heirloom varieties. In a qualitative survey, Bantle und Hamm [29] found a low level of knowledge about the loss of agrobiodiversity and its preservation through the use of heirloom varieties. This finding is supported by a study published in 2006 by Kleinhückelkotten et al. [9], who had not found an awareness of this topic in consumers' everyday lives either. In addition, there is a cognitive dissonance among consumers when they are confronted with the topic of agrobiodiversity loss, since in the subjective perception of consumers, the diversity in food retailing is increasing [9].

Knowledge about old varieties is often based on personal experience, e.g., whether an old variety has already been eaten by oneself. Gastronomy, media, and direct marketers can also serve as sources of information [29].

Consumers generally show a positive attitude towards heirloom varieties. They are associated with "nature/natural", "native/from here", "regional", "traditional", and an intensive taste. Modern varieties, on the other hand, are often rated negatively [29,30].

Consumers see the purchase of old varieties and their own cultivation, e.g., of fruit trees, as possible options for action to preserve old varieties [31]. Availability and price are seen as possible obstacles for their preservation [32].

Studies in other European countries have shown similar results. Notari and Ferencz [32] show a high interest in taste and regionality for traditional food in Hungary (especially tomatoes). A study on traditional maize carried out in Portugal shows a high

level of interest in taste and quality among consumers surveyed [33]. Meier and Öhen [15] show that, although the labeling as a traditional/old variety is not an important purchasing criterion, consumers do generally accept the idea of more diverse varieties (here farmers' varieties) and are (hypothetically) willing to pay more for them.

In the US, heirloom vegetables have a longer tradition than in many European countries and can be classified as a status symbol for conscious consumers [34]. Shoppers often declare taste as an important factor when buying heirlooms. However, a study by Joseph et al. shows that consumers value visual appeal and a greater selection variety over taste when buying heirloom varieties [35].

*2.4. Characteristics of Potential Target Groups for Heirloom Vegetable Varieties in Germany*

In Germany, an exploratory study by Kleinhückelkotten et al. [9] characterised potential target groups for biodiversity-enhancing products. According to the authors, two different target groups are particularly relevant: first, conscious organic shoppers, who are often characterised by a high income and an academic background. In Germany, about 23% of the population can be described as conscious regular consumers of organic products and 11% as convinced intensive consumers of organic products, indicating a high marketing potential [36].

There is a large number of studies on the purchasing preferences of organic consumers in Germany, most of them focusing on specific product groups (such as vegetables or meat) and relating to further benefits, such as regionality or animal welfare, e.g., as demonstrated in [37,38].

A study by Meyer-Höfer et al. [39] shows that the majority of convinced sustainable food consumers (including organic consumers) are female and see their personal purchasing decisions as an important factor influencing sustainable development. For German consumers, reasons for buying organic products include the desire for natural and healthy products, environmental protection, and a preference for local food. German organic consumers also value diversity, e.g., in regarding fruit and vegetables [40]. In addition, egoistic-hedonistic aspects, such as the desire for high-quality food and the pleasure of eating, are relevant [41]. Similar results regarding organic food consumption and attributes of organic food can be found for other European countries, e.g., as demonstrated in [42], and globally, e.g., as demonstrated in [43].

Secondly, the Traditional Clientele Is a Potential Target Group for Heirloom Vegetable Varieties. Representatives often have direct memories of heirloom varieties and aim to preserve them for reasons of nostalgia [9]. Lesser-known vegetables, such as parsnip or mangold, have experienced a boom in Germany in the past 10 years. Consumers appreciating these products may also be a relevant target group for heirloom varieties of more common vegetables, such as carrots or radishes [44].

## 3. Theoretical Framework to Classify Consumers' Purchasing Motives

There are various explanatory constructs for consumer behaviour and environmentally sound behaviour pointing out emotions and attitudes as important components in purchasing decisions [45,46].

In this paper, we focus on these attitudinal aspects using Stern's Value Belief Norm Theory [47] and Steg's Integrated Framework for Encouraging Pro Environmental Behaviour [48] to evaluate the collected data and to describe consumers' motivations to buy heirloom vegetable varieties.

Stern [47] states that environmentally conscious, private behaviour—such as individual shopping behaviour—is based on biospheric, altruistic, and egoistic values.

Steg et al. [48] use a different approach while working with similar value dispositions as Stern [47]. According to their Integrated Framework for Encouraging Pro Environmental Behaviour [48], environmental behaviour is based on three different types of motivations: hedonic goals, gain goals, and normative goals. When one or several of these goals are triggered, they activate further cognitive processes in the purchasing decision, leading to

an observable action (i.e., purchase of a given product). The underlying values set one or several of these motivations in focus. Hence, they propose the following:

- Altruistic and biospheric values affect the accessibility of normative motivations. Hence, consumers focus on what ought to be done to safeguard the environment, e.g., they buy heirloom varieties because of their value to agrobiodiversity.
- Egoistic values affect the accessibility of gain motivation. Here, consumers focus on their own resources, such as money or status, e.g., they buy heirloom varieties because doing so helps them gain approval from their peers.
- Hedonic values affect the accessibility of hedonic motivations, i.e., consumers try to seek pleasure or avoid efforts. They may buy heirloom varieties because they are interested in their special taste [48].

In this study, we combine these two motivational approaches and apply them in our analysis. Altruistic and biospheric motivations are summarized as self-transcendent purchasing motives, whereas hedonic and egoistic motivations are summarized as self-enhancement purchasing motives [48]. The VBN theory proves to be particularly suitable for explaining basic behavioural change in relation to more environmentally friendly behaviour, which includes individual shopping behaviour [47,49,50]. Similar theoretical approaches have already been used in other studies on sustainable purchasing behaviours in the food sector [51–53].

We apply both types of motivation in the context of heirloom varieties. Examples for both motivational factors, which have been derived from the existing literature [8–10,28,29,33,54], are depicted in Table 1.

**Table 1.** Motives to purchase heirloom vegetable varieties.

| Purchase Motivation | Examples |
|---|---|
| **Self-enhancing** <br> purchase of old varieties for one's own benefit or to improve one's own feelings | taste, attractive appearance, new recipes, health benefits |
| **Self-transcending** <br> purchase of old varieties to protect the environment for other people and for one's own sake | diversity, regionality, the adaptability of old varieties to climate change, benefits for horticulturist |

Existing studies show that egoistic-hedonic motivations typically correlate negatively with pro-environmental beliefs, attitudes, and behaviours [34,48].

Based on these two motivational factors and their influences on specific goals, Steg et al. [48] suggest two strategies to encourage pro-environmental behaviour:

- Reduce the conflict between gain/hedonic motivations and normative ones, e.g., by increasing the perceived personal benefits of an action.
- Strengthen the normative motivation and thereby weaken the other two, e.g., by putting a greater emphasis on the (positive) environmental outcome of an action.

## 4. Material and Methods

### 4.1. Focus Group Discussions

Data were generated from three focus group discussions in Berlin in May and June 2018. Focus groups can be defined as semi-structured group interviews. They are often used in marketing contexts. In focus groups it is intended that there are discussions between the participants, which then allow for a deeper understanding of the study subject than in individual discussions [55,56]. We chose Berlin as the study area, as it is one of the largest markets for organic food products in Europe. The city comprises different organic food distributors and retailers including weekly farmers markets, organic supermarkets, and market halls [57].

The recruiting for the group discussions was conducted via leaflets in organic supermarkets, newsletters, and social media channels of independent organic shops and

non-governmental organisations. The leaflet served as the main recruiting tool online and offline. It indicated that the focus of the discussions would be on heirloom vegetable varieties. In addition, a snowball system was used, as we asked potential participants to spread the call to other interested people in their social surroundings.

The participants had to contact the researchers themselves to participate in the discussion. This required a high level of motivation to participate and a high interest in the research topic. There was a risk that the interviewed group remained homogeneous [58].

However, the recruitment process was chosen to ensure that participants had some prior knowledge on the topic to assess their attitudes towards heirloom varieties. Participants were selected according to set criteria based on the analysis of the potential target groups. The participants agreed to a scientific evaluation of their contributions as well as to the publication of their statements in an anonymised form in a scientific context.

In total, 15 participants took part in three focus group discussions, which was sufficient for thematic saturation among organic consumers. One group discussion comprised 4 participants, one 5, and one 6, which lie within literature recommendations for small focus groups [56]. Each participant received a reward of 15€ after participating in the discussion.

The demographics of the participants show that the groups were quite homogenous. The majority of the participants were women with a high educational level, though well mixed in age. Two thirds of the participants stated that they preferred to go grocery shopping in organic food stores. Hence, we classified them as conscious regular consumers of organic products. Key information on the participants' demographic background is presented in Table 2.

**Table 2.** Participants in the focus group discussions to heirloom vegetable varieties (*n* = 15).

| Demographics | Description of Participants |
| --- | --- |
| Gender | Female: 13<br>Male: 2 |
| Age range | >25: 3<br>26–40: 6<br>41–55: 4<br>56–70: 2<br><70: 0 |
| Educational level * | Low: 0<br>Medium: 3<br>Superior: 12 |
| Usual place of vegetable Purchase (Multiple answers possible) | Conventional supermarkets: 5 |
| | Organic supermarkets: 10 |
| | Others: 12 |

* Low: no degree; Medium: secondary school or vocational training; Superior: High school diploma or university degree.

### 4.2. Discussion Topics and Procedure

The moderator used a semi structured interview guide within the focus group discussions. The precise wording and sequence of questions were not fixed and could be adapted according to the group [59,60]. Some questions contained hypothetical aspects, as heirloom varieties are currently only sold in direct marketing and not in organic supermarkets where the majority of the participants do their grocery shopping. Five main topics were covered in each discussion:

- General consumer focus when purchasing vegetables;
- Knowledge of and attitudes towards heirloom vegetable varieties;
- Evaluation of existing communication approaches for heirloom vegetable varieties;

- Information requests for heirloom varieties;
- Existing concerns towards heirloom varieties.

The interview guide was pretested with a focus group of students of the Eberswalde University for Sustainable Development, after which some minor corrections were made.

In total, we conducted three interviews (excluding the pretest). They lasted 90 min on average. The interviews were audio recorded and later transcribed verbatim.

### 4.3. Tested Communication Approaches

In the focus groups, we examined four communication approaches for their suitability to communicate the benefits of heirloom vegetable varieties to consumers. We used existing communication material to be close to a real world setting, but we altered the material slightly to fit the regional context (Table 3). The communication approaches were presented to the participants as small leaflets. Each leaflet included a logo, a key message, and an additional text element. Questions on the leaflets included consumers' general opinions on the design, their trustworthiness, and appealing arguments. We did not control for design and colours.

**Table 3.** End-consumer communication tools for heirloom varieties used in the focus group discussions comprising a logo and key message (xxx- strong focus on egoistic-hedonic or altruistic-biospheric motivations; x- little focus).

| Organisation | Logo and Key Messages | Self-Enhancement Motivation | Self-Transcendent Motivation |
|---|---|---|---|
| 'Pro Specie Rara' Swiss foundation for the cultural and genetic diversity of plants and animals |  Rediscover heirloom varieties! If you want to protect heirloom varieties, you gotta eat them! | XXX Taste, appearance | XXX Cultural heritage, adaptation capacity |
| 'Ostmost' Berlin-based company producing juice and cider from orchard meadows |  Flavour boom from orchard meadows. Drink and rebel! | XXX Taste, hedonic product design | XX Diversity, ecosystem functions |
| Vielfalt schmeckt' ('Diversity Tastes Delicious') Project of 'Pro Specie Rara' Germany and a regional natural food wholesaler to promote heirloom varieties |  Save - Preserve - Spread A new future for the crops on the „RED LIST" | X Taste | XXX Adaptation capacity, Diversity |

**Table 3.** *Cont.*

| Organisation | Logo and Key Messages | Self-Enhancement Motivation | Self-Transcendent Motivation |
|---|---|---|---|
| Wie Früher ('Like Old Times'): Own brand of the Austrian supermarket chain store 'Spar'. The claim was adapted to the regional context. | *Natural delicacies from Berlin and Brandenburg* wie früher | XX Taste, quality, exclusivity | X Regionality |

The communication approaches were chosen based on a literature analysis considering the proposed theoretical framework. They were selected to address different purchasing motives and thereby presented the participants with a wide range of communication approaches (Table 3):

- *'Like Old Times'* appeals to self-enhancement purchasing motives only.
- *'Ostmost'* addresses self-enhancement as well as self-transcendent purchasing motives, such as the conservation of heirloom varieties.
- *'Diversity Tastes Delicious'* focuses on self-enhancement purchasing motives and features the *'Red List'* in its central claim, pointing out the possible loss of heirloom varieties.
- *'Pro Specie Rara'* addresses self-enhancement as well as self-transcendent purchasing motives.

We mainly chose communication approaches that were not present in Berlin at the time of the survey. Thus, the participants' statements focused on the slogans and text elements instead of on the brands themselves. The only exception was the "Ostmost" claim, as there was no other existing brand reflecting that specific communication approach.

*4.4. Analysis*

The evaluation of the collected data gained in the discussions is based on a qualitative content analysis [61]. We used the software MAXQDA and evaluated the material according to thematic categories (see Supplementary Materials). Main categories were set deductively, based on the interview guide developed. We built the subcategories inductively, based on the collected material. Complete statements served as the unit of analysis. Several passages were multi-coded as the motivations examined were mentioned in context with multiple categories.

**5. Results**

*5.1. Knowledge and Attitudes towards Heirloom Vegetable Varieties*

Although the participating consumers had a prior interest in heirloom varieties due to the recruitment process, their knowledge on this topic was fairly low. Participants with primary experience, e.g., from harvesting assignments in Community-Supported Agriculture or as owners of a home garden, were more familiar with the topics discussed. Asked to define heirloom vegetable varieties, consumers mentioned species with little market presence, such as Jerusalem artichoke, cabbage, or black radish. They could only specify very few specific names of different heirloom varieties (e.g., the potato variety 'Bamberger Hörnchen', the tomato variety 'Harzfeuer'). Participants stated that they only occasionally pay attention to the variety they buy and therefore could not state whether they buy old or modern varieties. Most participants did not distinguish between the terms "species" and "variety".

Nevertheless, participants stated predominantly positive associations with heirloom varieties, such as an intense taste, high robustness, naturalness, and health-promoting properties.

In contrast, the participants viewed modern varieties with skepticism, as they were associated with a lower taste intensity and negative impacts on health, such as malnutri-

tion. In addition, the participants suspected that modern varieties were heavily bred for appearance, rapid growth, uniformity, and size.

The interviewed consumers also raised some concerns about heirloom varieties. They assigned a higher price to heirloom vegetable varieties than to modern ones. Reasons included the expected lower shelf life of old varieties and the additional expenditure for horticulturists. In addition, heirloom varieties were attributed a lower yield and a higher susceptibility to pests. Moreover, the participants raised the question whether heirloom varieties were still adapted to today's climate. In addition, they discussed the issue of digestibility. Another important point was their personal lack of knowledge about the correct preparation of heirloom varieties for cooking.

*5.2. Evaluation of Communication Approaches*

The leaflets featuring existing communication approaches for heirloom vegetable varieties were evaluated on three levels. First, it was recorded how many statements were made on the respective communication tool. Secondly, these statements were divided into positive, negative, and neutral statements. Third, statements that could be assigned to a specific purchase-motivated orientations were recorded and evaluated separately (see Section 4.4).

Within the three discussions, far more positive than negative statements were made about the leaflets. However, none of the communication approaches tested convinced all consumers.

'*Diversity Tastes Delicious*' and '*Ostmost*' were discussed most often and assessed predominantly positively. This can be explained by individual preferences regarding the graphic elements on the leaflets as well as the content presented. In the following section, we focus on the content level.

On the leaflet '*Ostmost*', a comparison between the diversity of orchard meadows and tropical rainforests was seen particularly positively, as consumers were surprised by this information and could personally relate to orchard meadows. Moreover, the leaflet included specific variety names which, were appreciated by the customers interviewed. The sentence "*A difference you can taste*" was also predominantly rated positively.

The leaflet '*Diversity Tastes Delicious*' included the sentence "*conservation through utilisation*" which was mentioned several times with positive connotations. Additionally, it featured the '*Red List*', which received a mixed evaluation. The '*Red List*' was already known to participants from nature conservation, which conferred a trustworthy image for some of the interviewed consumers. Some participants claimed that to them the term '*Red List*' refers predominantly to endangered animal species. The term was new to the participants in the context of cultivated crops. However, it was correctly interpreted in the given context, as the respondents understood the '*Red List*' as a call for the preservation of heirloom varieties. Some interviewees criticised the term '*Red List*' as too emotionalising. One participant stated:

> "*Yes, 'Red List' [ . . . ], but in my opinion it is a bit too dramatic. The consumer might just want to consume and not hear about any threats*". (P3, FGD3)

The leaflet of '*Pro Specie Rara*' was evaluated inconsistently. No clear tendency could be determined. The text was described as more informative than the comparative ones. One participant stated:

> "*But they somehow explain the complexity of the system and the reason why it is good to eat such heirloom varieties and thus support the cultivation*". (P5, FGD1)

The key message "*If you want to protect old varieties, you gotta eat them*" was also positively evaluated. The text included a comparison of heirloom varieties with cultural assets, which was well received. However, the respondents criticised the Latin name '*Pro Specie Rara*' as too exclusive and only understandable to a rather highly educated clientele.

The leaflet '*Like Old Times*' was predominantly negatively rated. It was conceived as inconvincible. Only one participant saw the reference to regional origin and exclusive products positively.

### 5.3. Information Requests

Consumers were most interested in taste and health-promoting properties of heirloom varieties. The participants in the focus group discussions assumed that heirloom vegetable varieties were healthier than their modern counterparts. The variety name, the characteristics of heirloom varieties, the origin, and the age of the variety were also of interest.

In addition, some participants were curious about difficulties for horticulturists who cultivate heirloom varieties and about the benefits they derive from them. One participant stated:

> *"I would also be interested to know why this variety is being re-introduced into the system. Is that one special in taste? Is it particularly resistant to any pests against a virus or bacterium?" (P2, FGD3)*

The participants suggested to include conservation organisations enjoying a high level of trust in communicating the benefits of heirloom vegetables.

Furthermore, the participants pointed out where one could learn about heirloom varieties at the Point of Sale. This included the price tag, recipe suggestions, leaflets, posters, and information in customer magazines.

### 5.4. Purchasing Motivations

In total, 84 statements made by the participants can be assigned to one of the purchasing motivations analysed in this study. A total of 47 statements can be allocated to self-enhancing motivations and 37 to self-transcendent motivations.

### 5.4.1. Self-Enhancement Motivation

Statements appealing to egoistic-hedonic motivations mainly contained assumptions about a more intensive and better taste of old vegetable varieties, which can be demonstrated by the following statement:

> *"I would also be willing to pay more [for heirloom vegetables] if it really is a taste experience". (P1, FGD 3)*

In addition, heirloom varieties were often attributed with health-promoting properties, and the participants expected more secondary plant compounds in old varieties.

Hedonic-egoistic statements were predominantly recorded when participants spoke about their information requests.

### 5.4.2. Self-Transcendent Motivations

Some participants showed an intrinsic motivation to preserve heirloom varieties. For them, diversity itself was a good worth protecting. They also mentioned its contribution to global nutrition and the stability of ecosystems. One consumer pointed out:

> *"So for me, I see diversity as the only solution for everything [..]. Also worldwide nutrition". (P4, FGD 3)*

The interviewed consumers associated heirloom varieties with an added value for horticulturists, e.g., better income generation. Many positive characteristics were attributed to heirloom varieties. Those included pest resistance, adaptability to climate change, or suitability for regional soil conditions. The participants also saw a benefit for the entire ecosystem. These statements suggest an altruistic-biospheric interest in heirloom varieties. The focus on future challenges, and thus the contribution of heirloom varieties to intergenerational justice, was emphasised on two occasions. Two participants also touched on the topic of regional origin.

Statements suggesting an altruistic-biospheric orientation were predominantly recorded when the participants discussed the presented communication approaches.

## 6. Discussion

Overall, this study shows that organic consumers have a positive image of heirloom vegetable varieties. Accordingly, old vegetable varieties have a certain market potential that can be increased with the appropriate communication concept.

Existing consumer trends, such as indulgence, quality, or variety seeking, support the market introduction of heirloom varieties [62].

From our results some important implications for developing a communication concept can be drawn. Despite the previous interest of the participants in the subject of heirloom varieties, their knowledge of the subject was rather limited, suggesting that the choice of variety is not a decisive factor when buying vegetables. Accordingly, when communicating about heirloom varieties, basic information (e.g., "what is a variety?") should be included.

In addition, some contradictory statements were made by the participants: heirloom varieties were expected to have high robustness on the one hand and low pest resistance on the other. This indicates that the statements on vegetable varieties were association-driven and not knowledge-based. Those results are in line with the findings of existing studies [28,29]. Thus, communication of the added value of old varieties should tie in with a relatively low level of consumer knowledge.

With regard to the content of the leaflets, consumers were most interested in the taste of heirloom vegetable varieties and their potential health-promoting properties. This result is also reflected in other studies [63,64]. Raised assumptions about health-promoting properties can be ascribed to a variety of internet blogs and publications in popular science (e.g., [65]) as well as to positive health messages for heirloom grain varieties (e.g., low gluten content) [8]. Existing studies from Portugal and Hungary show that traditional varieties are considered healthier and more tasteful than their modern counterparts. Regional origin, traditions, and childhood memories are favourable factors for their good reputation [32,33,66]. Therefore, heirloom varieties can be particularly relevant for organic consumers, as these items are also important for buying organic products (e.g., regionality, traditions), e.g., as demonstrated in [37].

Moreover, some participants lacked trust in modern agricultural systems and assumed that heirloom varieties were more natural and healthy than modern varieties [66].

Building on existing communication approaches and on the opinions expressed in our study, an optimistic narrative should be developed. It is noticeable that the leaflets of 'Ostmost', 'Pro Specie Rara', and 'Like Old Times' convey optimistic images and emphasise the advantages of heirloom vegetables, whereas 'Diversity Tastes Delicious' highlights the potential danger of their extinction. It has been shown that positively framed messages generally elicit favourable responses for a given target group [67]. Hence, one can expect that positively framed communication approaches are more motivating for consumers to buy heirloom vegetable varieties. However, the example of 'Like Old Times' also conveys that this effect should not be overstretched. While using positive frames, it almost exclusively appeals to self-enhancing motivations and hence loses credibility. Kareklas et al. [68] suggest that advertisements combining egoistic and altruistic claims produce more favourable responses towards organic brands than purely egoistic or altruistic-focused advertisements.

Concerning the 'Red List' as a potential communication approach for heirloom vegetable varieties, the obtained data do not reveal a clear trend. Janssen and Hamm [69] show that the familiarity of a communication scheme can have positive effects on its popularity. As consumers knew the 'Red List' from other contexts, one could argue that it is a useful instrument to communicate the benefits of heirloom vegetable varieties. However, one must expect that consumers have a low level of cognitive involvement at the Point of Sale [70]. The required cognitive effort to associate the 'Red List' with threatened vegetables and not with threatened animal species is particularly high, which can reduce its efficiency in real-life settings. Therefore, we cannot give a recommendation for the 'Red List' as a possible communication approach for heirloom varieties.

Concerning consumers' information needs which need to be addressed, it was noticeable that more statements showing self-transcendent purchasing motives were made in the context of describing and evaluating the presented communication approaches, whereas more self-enhancement motives were stated when the consumers discussed their information requests towards heirloom varieties. Due to the exploratory character of the study, this slight overrepresentation should not be overestimated. Nevertheless, this tendency indicates that the tested communication approaches for heirloom vegetable varieties do not sufficiently appeal to the egoistic-hedonic motivations expressed by the consumer's information demands. Existing studies on sustainable food consumption and production support this tendency, as they indicate a higher importance of egoistic-hedonic motives. Birch et al. [53] show that the influence of egoistic motives—such as health benefits and food safety—are more important than altruistic motives—such as environmental friendliness—when buying local food. Studies on members of Community-Supported Agriculture show a balance between egoistic and altruistic goals with a slight tendency towards egoistic ones [52,71].

Finally, looking at the theoretical framework used in this paper and at the different suggestions to encourage pro-environmental behaviour made by Steg et al. [48], one should focus on their first proposed strategy: reduce the conflict between gain/hedonic motivations and normative ones. This can be conducted by highlighting personal benefits, such as communicating the taste of heirloom varieties and developing recipe suggestions. The normative goal can be strengthened by pointing out the positive environmental outcomes of preserving heirloom varieties by using specific examples, e.g., comparing the diversity of orchard meadows and tropical rainforests, as in the example of 'Ostmost.'

## 7. Conclusions and Limitations

The aim of this study was to examine consumer attitudes and information requests towards heirloom vegetable varieties, determine the purchasing motivations which need to be addressed to efficiently transport the benefits of biodiversity-enhancing products, and to evaluate existing communication approaches for these products.

As a theoretical contribution to consumer research on the issue of the protection of plant-genetic resources, our data suggest that the general attitude towards heirloom vegetable varieties is positive, although some concerns have been raised (e.g., higher price, lower shelf life).

Furthermore, the results show that the tested communication approaches strongly focus on self-transcendent motivations, whereas consumers' information requests place a stronger emphasis on self-enhancing ones. Hence, one should focus on the latter when communicating the added value of heirloom vegetable varieties. This can be conducted by highlighting the special taste of heirloom varieties, e.g., by providing recipe suggestions. If altruistic-biospheric motivations are addressed, the images and frames used should be positive, highlighting the advantages of heirloom varieties instead of their possible extinction in a manner that is as specific and consumer-oriented as possible.

As part of value chain development, the results obtained in this paper are a vital component for a communication concept for heirloom vegetables in organic food retailing. This is an important step to make heirloom vegetables accessible to a greater share of consumers. Although the study was conducted in Germany, the results may also be applied in other contexts in which a marketing differentiation of varieties is made (e.g., farmers varieties, organic breeding, open-source varieties) [66]. As citizens from different countries vary in their assessments of the added value of organic farming [15,72], it can be interesting to conduct similar studies in other countries with well-developed organic markets (e.g., the United States, France, and China).

However, we want to point out some limitations of this study. Due to the sampling process, only organic consumers who already had a high interest in the topic of heirloom vegetables took part in the focus group discussions. In addition, the rather traditional target group was not addressed in the focus group discussions. The survey had a strong

gender bias, as only two male participants took part in the discussions. However, as grocery shopping is more often performed by females than by males, the results are still applicable [73]. The study was conducted in an urban environment, and thus the results may not be applicable to a rural population.

In future research it is therefore advisable to re-examine the results in a quantitative study with a greater and more heterogeneous sample. The study was conducted prior to the COVID-19 pandemic and hence can be repeated, as the valuation of consumers towards heirloom vegetable varieties may have changed.

A general challenge of focus group discussions is the possibility of socially desirable answers [55], which was not addressed in this paper.

**Supplementary Materials:** The following supporting information can be downloaded at: https://www.mdpi.com/article/10.3390/su14074068/s1, MAXQDA software.

**Author Contributions:** Conceptualization, J.L. and C.B.; methodology, J.L. and C.B.; formal analysis, J.L.; data curation, J.L.; writing—original draft preparation, J.L.; writing—review and editing, J.L. and C.B.; supervision, C.B.; project administration, C.B.; funding acquisition, C.B. All authors have read and agreed to the published version of the manuscript.

**Funding:** This work was supported by funds from the Federal Ministry of Food and Agriculture (BMEL) based on a decision by the parliament of the Federal Republic of Germany via the Federal Office for Agriculture and Food (BLE) under the Federal Programme for Ecological Farming and Other Forms of Sustainable Agriculture under Grant no 2815NA179.

**Institutional Review Board Statement:** The focus groups cited in the article were conducted using the informed consent principle. Before his or her participation in the focus group, each person filled out a form stating his or her informed consent. The participants agreed to the publication of their anonymised statements within the research project and for academic publications. Participation in the focus groups was voluntary. Participants had to contact the researchers themselves, as the invitation to the focus group discussions was distributed via public notices and social media. Only healthy adults over the age of 18 were included in the study. This procedure complies with the guidelines of the Deutschen Forschungsgemeinschaft (DFG) for Humanities and Social Sciences. https://www.dfg.de/en/research_funding/faq/faq_humanities_social_science/index.html accessed on 1 February 2022.

**Informed Consent Statement:** Informed consent was obtained from all subjects involved in the study.

**Data Availability Statement:** The data presented in this study are available on request from the corresponding author.

**Conflicts of Interest:** The authors declare no conflict of interest.

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
