# Peer review of "“For More Diversity, Better Taste and My Own Health” Exploring Organic Consumers’ Purchasing Motives for Heirloom Vegetable Varieties"

_sustainability, doi:10.3390/su14074068_

Round 1
Reviewer 1 Report
First of all, let me congratulate the author on a very interesting study. Overall, this paper is well written and structured. The model seems to me nice and the results are quite good even if there some limitations that the author is aware about.
The survey data and statistical analysis are well presented and the results could be interesting but the article should have a review of literature and a discussion more related to the activity, more structured, in order to justify the methodological choices and to show the contribution of the study.
Some past researchers have been studied the issues. Comparing the previous studies, what is major implication for academicians and enterprisers? Author should be discussed in discussion and implications section. I recommend acceptance with major revisions.
Author Response
Dear Sir / Madam
Thanks a lot for your valuable feedback.
Based on your suggestions, we have restructured our discussion section and oriented it more clearly towards the implications for enterprises. These include concrete proposals for the design of a communication concept for heirloom vegetable varieties. Those are based on the results, the theory used and additional references in the discussion (Section 6).
In addition, we have adjusted our literature review to emphasise the focus of this paper on organic consumers (Section 2.5).
Reviewer 2 Report
This is a good quality manuscript. The research problem is relevant and important. The applied methodology is correct. The results are presented clearly.
Please refer to:
Bryła P., Organic food consumption in Poland: Motives and barriers, Appetite, 2016, Vol. 105, pp. 737-746. https://doi.org/10.1016/j.appet.2016.07.012.
Bairagi, S., Custodio, M.C., Durand-Morat, A. et al. Preserving cultural heritage through the valorization of Cordillera heirloom rice in the Philippines. Agric Hum Values 38, 257–270 (2021). https://doi.org/10.1007/s10460-020-10159-w.
line 167 - of potential
170 - who
222 - weaken
223 - emphasis on
268 - do
419 - varieties,
548 - socially
549 - [46],
Reference 39 - journal title
a
Author Response
Dear Sir / Madam
Thanks a lot for your valuable feedback and for taking your time to point out some language issues.
In the literature review, we have now placed a greater focus on organic consumers (in Germany). As a short overview on other European countries, we have taken into account the references you listed here:
Bryła P., Organic food consumption in Poland: Motives and barriers, Appetite, 2016, Vol. 105, pp. 737-746. https://doi.org/10.1016/j.appet.2016.07.012
Moreover, we have included the article:
Bairagi, S., Custodio, M.C., Durand-Morat, A. et al. Preserving cultural heritage through the valorization of Cordillera heirloom rice in the Philippines. Agric Hum Values 38, 257–270 (2021). https://doi.org/10.1007/s10460-020-10159-w .
As our work has a strong focus on highly developed organic markets, we have not included the interesting results of this study in the discussion, but only used them in the introduction to point out the need for functioning marketing strategies.
Reviewer 3 Report
I think that the paper is quite well written and that it has a relevant contribution to the literature.
I would suggest that the authors make a distinct section for the literature review which should come after the introduction.
The literature review seems now to be split on to many subsections. I think that this is not really proper.
The relationship between customer value co-creation behavior, consumer perceptions, and acquisition decisions as regards organic consumers’ purchasing motives has not been covered, and thus such recent sources should be cited: Meilhan, D. (2019). “Customer Value Co-Creation Behavior in the Online Platform Economy,” Journal of Self-Governance and Management Economics 7(1): 19–24. doi: 10.22381/JSME7120193. Watson, R., and Popescu, G. H. (2021). “Will the COVID-19 Pandemic Lead to Long-Term Consumer Perceptions, Behavioral Intentions, and Acquisition Decisions?,” Economics, Management, and Financial Markets 16(4): 70–83. doi: 10.22381/emfm16420215. Graessley, S., Horak, J., Kovacova, M., Valaskova, K., and Poliak, M. (2019). “Consumer Attitudes and Behaviors in the Technology-Driven Sharing Economy: Motivations for Participating in Collaborative Consumption,” Journal of Self-Governance and Management Economics 7(1): 25–30. doi: 10.22381/JSME7120194. The relationship between consumer choice, cognitive attitudes, and purchasing habits as regards organic consumers’ purchasing motives has not been covered, and thus such recent sources should be cited: Mirică (Dumitrescu), C.-O. (2019). “The Behavioral Economics of Decision Making: Explaining Consumer Choice in Terms of Neural Events,” Economics, Management, and Financial Markets 14(1): 16–20. doi: 10.22381/EMFM14120192. Rydell, L., and Kucera, J. (2021). “Cognitive Attitudes, Behavioral Choices, and Purchasing Habits during the COVID-19 Pandemic,” Journal of Self-Governance and Management Economics 9(4): 35–47. doi: 10.22381/jsme9420213. Drugău-Constantin, A. (2019). “Is Consumer Cognition Reducible to Neurophysiological Functioning?,” Economics, Management, and Financial Markets 14(1): 9–14. doi: 10.22381/EMFM14120191. The authors rely also on too many very short paragraphs consisting of only 1-2 phrases. The methodology is quite nicely described. The results are properly presented. In the discussions section I would expect more comparisons between own results and previous findings of the literature in order to highlight the novelty of the paper. The conclusions MUST consist of 4 paragraphs: theoretical contributions managerial implications limitations future research perspectivesAuthor Response
Dear Sir / Madam
Thanks a lot for your valuable feedback. We have restructured our paper according to your suggestions. Our new literature review (section 2) now includes a subsection on the importance of plant genetic resources, heirloom varieties, consumer attitudes towards heirloom varieties as well as a characterisation of the potential target group with an emphasis on organic consumers. We restructured our discussion to highlight the implication our paper has for marketing of heirloom varieties. Furthermore, we restructured the conclusion (section 7) pointing out theoretical contributions of our findings to consumer research, managerial implications for marketing heirloom varieties, limitations the chosen method and future research perspectives.
As our theoretical framework focusses on a psychological approach (Value Belief Norm Theory, Integrated Framework for Encouraging Pro Environmental Behaviour) we did not include the suggested articles by Meilhan (2019). Moreover, the suggested articles on neuroscience were beyond the scope of our theoretical framework. Nevertheless, we thank you for the suggesting this approach which will be interesting in further research in this topic.
Furthermore, we excluded potential changes in consumer attitude towards heirloom varieties during the Covid 19 pandemic as the interviews were conducted in 2018 (see your suggestion Watson et al 2021).
Round 2
Reviewer 1 Report
No more comments